# Contamination Level, Distribution Characteristics, and Ecotoxicity of Tetrabromobisphenol A in Water and Sediment from Weihe River Basin, China

**DOI:** 10.3390/ijerph17113750

**Published:** 2020-05-26

**Authors:** Xueli Wang, Chenyang Li, Xiaoyu Yuan, Shengke Yang

**Affiliations:** 1Key Laboratory of Subsurface Hydrology and Ecological Effects in Arid Region, Ministry of Education, Chang’an University, Xi’an 710054, China; lcy90170804@126.com (C.L.); 15129037687@163.com (X.Y.); ysk110@126.com (S.Y.); 2School of Water and Environment, Chang’an University, Xi’an 710054, China

**Keywords:** tetrabromobisphenol A, sediment, water, Weihe River Basin, distribution, risk assessment

## Abstract

Tetrabromobisphenol A (TBBPA) is a brominated flame retardant, which is widely present in the various environmental and biological media. The knowledge on the contamination of TBBPA in Weihe River Basin is still limited. In order to know the pollution level and distribution of tetrabromobisphenol A (TBBPA) in the Weihe River Basin, a total of 34 sediment samples and 36 water samples were collected from the main stream and tributaries of the WeiHe River Basin, and the concentration of TBBPA in the samples was analyzed by high-performance liquid chromatography–electrospray ionization–mass spectrometry (HPLC-ESI-MS). The detection frequency of TBBPA in sediments and water samples was 61.8% and 27.8%, respectively; the TBBPA concentrations in sediments and water samples were in the range of not detected (N.D.)–3.889 ng/g (mean value of 0.283 ng/g) and N.D—12.279 ng/L (mean value of 0.937 ng/L), respectively. Compared with other areas in China, the residues of TBBPA in the Weihe River Basin were at a relatively low level. The spatial distributions of TBBPA in surface sediments and water indicated that the local point-input was their major source. This is related to the proximity of some sampling sites to industrial areas and domestic sewage discharge areas. The insignificant correlation between TBBPA and total organic carbon (TOC) indicated that TBBPA in sediments is not only influenced by TOC but also affected by atmosphere and land input, wet deposition, and long-distance transmission. The potential risks posed by TBBPA in water and sediment were characterized using the risk quotient (RQ) method. The calculated RQ for TBBPA was less than 0.01, showing that the ecological risk due to TBBPA was quite low for aquatic organisms.

## 1. Introduction

Tetrabromobisphenol A (TBBPA) is the most widely used brominated flame retardant in the world, accounting for more than 60% of the brominated flame retardant market, and is extensively used in printed circuit board, paper, textile, and other industrial products [1]. It was demonstrated in studies that TBBPA has strong immunotoxicity, endocrine toxicity, and the characteristics of persistent pollutants such as long-distance migration, bioaccumulation, and toxicity [2,3]. TBBPA continues to enter the environment in the process of production, use, and disposal of related products. Therefore, the harm of TBBPA to the environment and human body has been widely considered, and it has been detected in various biological and environmental matrices including soil, atmosphere, water, sediment, animals, and plants [4,5,6,7].

Recent studies indicate that there are a number of pathways through which TBBPA can enter the environment, including release from consumer products, emissions from production processes, and leaching from treatment sites. As a reactive flame retardant, TBBPA does not migrate from the product. However, as an additive flame retardant, TBBPA can be released from the product and enter indoor air and dust [8]. When TBBPA was used as an additive flame retardant, it was mainly used in the outer casing of electronic products such as televisions and computers. It has been found that the main sources of TBBPA in indoor air and dust were old electronic devices and electrical appliances, especially televisions, computers, etc. [9]. Therefore, in places where the population density is high, and the activity area is strong, the large-scale use of electronic products containing TBBPA seems to be a great contributor to its concentration increase in the environment.

Weihe River, the largest tributary of the Yellow River, mainly flows through Tianshui, Baoji, Xianyang, Xi’an, and Weinan. Millions of people live along the Weihe River Basin, and there are many chemical, electronics manufacturing, and textile enterprises in the Weihe River Basin. Rapid population increases and economic development have led to large quantities of anthropogenic contaminants entering the Weihe River from primary sources in runoff, in industrial and domestic effluent, and through atmospheric deposition. Previous studies have reported that organic-chlorine pesticides [10] and polycyclic aromatic hydrocarbons [11] were detected in the environmental media of Weihe River Basin. However, few studies have been focused on flame retardant pollutants, and the related research on TBBPA has not yet been carried out. Therefore, investigation and research on new persistent organic pollutants (POPs) such as TBBPA has become very urgent and necessary. In this study, we selected the Weihe River Basin of the Shaanxi section as the research area to analyze the concentration level, spatial distribution characteristics, and sources of TBBPA in sediment and suspended solids, which can provide the basic data for the assessment of environmental occurrence in the Weihe River Basin. The obtained results could provide data support for the global inventory of TBBPA and provide data information for future work on local risk assessment and pollution control.

## 2. Materials and Methods

### 2.1. Chemicals

Native TBBPA standard solution was purchased from AccuStandard (New Haven, CT, USA). TBBPA standard was obtained from Wellington Laboratories (Guelph, ON, Canada). Pesticide residue analysis grade acetone, dichloromethane, and n-hexane were purchased from Honeywell (Morris Plains, NJ, USA). High-performance liquid chromatography-grade acetonitrile and methanol were purchased from Fisher (Hampton, NH, USA).

### 2.2. Sample Collection

In June 2017, a total of 36 sampling points were set in the mainstream and tributaries of the Weihe River, 36 water samples and 34 sediment samples were collected. The location of the sampling sites is shown in Figure 1. Water samples were collected in brown glass bottles with a fine grinding mouth. The sample was filled in the bottle ensuring that it did not contain any bubbles. Then, several drops of concentrated hydrochloric acid were added to the bottles to make the pH value of the water sample less than or equal to 2. Sediment samples were placed in a clean self-sealing aluminum/polyethylene bag with a zip closure to keep the samples at a constant temperature of −18 °C. After the samples were transported back to the laboratory, the water sample pretreatment was completed within one week, and the sediment sample was completed within two weeks.

### 2.3. Sample Extraction and Cleanup

Pretreatment of sediment sample: The sample was ground through a 60 mesh sieve. A 10 g aliquot of a sample was added into a quartz filter paper cylinder with 2 g of copper granules, and then a substitute solution of ^13^C_12_-TBBPA 50 μL (1 μg/mL) was added into the sample. Then, it was Soxhlet extracted with a mixture of n-hexane:dichloromethane (1:1, v/v) for 16 h. Anhydrous sodium sulfate was added for dehydration, and then the extract was evaporated to 3.0 mL using a rotary evaporator. The silica gel column was activated with 50 mL of n-hexane, and 3.0 mL of extract was added to the silica gel column. The column was eluted with a 3/1 v/v mixture of hexane and dichloromethane, and the solvent was concentrated to near dryness. Next, 50 μL of ^13^C_12_-α-HBCD (1 μg/mL) was added, and then the solution was diluted to 1 mL with methanol. The samples were filtered through a 0.22 μm organic filter, and analyzed by HPLC-ESI-MS (Thermo Fisher Scientific TSQ Quantum Access MAX, USA).

Water sample treatment: 5 mL dichloromethane, 5 mL methanol, and 5 mL deionized water were used to activate a C18 (500 mg, 6 mL) solid-phase extraction column. Then, a 1000 mL water sample was added to 50 μL of the substitute TBBPA-^13^C_12_ solution (1 μg/mL), and then passed through an activated solid-phase extraction column at a flow rate of about 10 mL/min. The extraction column was eluted with 5 mL of deionized water, and the column was purged with nitrogen for 10 min. Following this, 5 mL of dichloromethane was used to elute the extraction column, and the extraction was received by the collection tube. Then, the solution was concentrated to near dryness, and the subsequent process was consistent with the sediment sample processing.

### 2.4. Instrumental Analysis

The TBBPA was measured using a C18 liquid chromatography column (2.1 × 150 mm i.d., 3.0 μm, Waters, USA). The column temperature was kept at 40 °C during an analytical run. The mobile phase flow rate was 250 μL/min. The injection volume was 10.0 μL. Three mobile phases were used, (A) water, (B) methanol, and (C) acetonitrile, and the flow rate was 0.2 mL/min. The mobile phase gradient elution procedure was as follows: it started with A/B/C 25/20/55 (v/v/v) and was ramped to A/B/C 10/20/70 (v/v/v) in 12.0 min, then changed to A/B/C 0/0/100 (v/v/v) in 0.2 min (maintained for 8 min), finally returned to A/B/C 25/20/55 (v/v/v), maintained for 9 min.

The mass spectrometer was operated in electrospray negative ionization mode. The triple quadrupole mass spectrometer was operated in selected reaction monitoring mode. The capillary spray voltage and capillary temperature were 3 kV and 230 °C, respectively. The sheath gas was nitrogen, the temperature was 310 °C, and the pressure was 28 psi. The auxiliary gas was also nitrogen, and the pressure was 5 psi. The quantified ions *m*/*z* 554.8 were monitored for ^13^C_12_-TBBPA. The scanning time was 250 ms and the tube lens offsets was 75 V.

The total organic carbon (TOC) content of each sediment sample was determined using a Vario TOC cube system. A 0.02 g aliquot of the dried sediment sample was loaded into the combustion cup. The sample was wetted with 3% (v/v) phosphoric acid and heated to 250 °C to remove inorganic carbon. The sample was then heated to 900 °C in the combustion house. The detection time was 6 min.

### 2.5. Quality Assurance and Quality Control

Isotope dilution method was used to detect the concentration of TBBPA in the present study. The three times signal-to-noise ratio found was taken as the detection limit of the instrument, and ten times signal-to-noise ratio was taken as the limit of quantification. The detection limit is 0.45 μg/L, and the quantitative limit is 1.5 μg/L. The recovery of sediment samples was 57.96–145.75%, and the recovery of water samples was 41.75–129.28%, which were in line with the national environmental protection standard (23–153%). No target compounds were detected in the blank experiments.

### 2.6. Ecological Risk Assessment

Toxicological data for TBBPA were obtained from the U.S. Environmental Protection Agency’s EPA ECOTOX Toxicology Database. The ecological risks presented by biotoxicity in the aquatic environment were evaluated using the risk quotient (RQ). The individual RQ was calculated as a ratio of the measured environmental concentration (MEC) and predicted no-effect concentration (PNEC) (RQ = MEC/PNEC). The PNEC is computed by constructing the Species Sensitivity Distribution (SSD) curve. The SSD curve was constructed using chronic toxicological data of aquatic organisms, and the curve fitting was performed using SigmaPlot 14.0 software (SYSTAT, San Jose, CA, USA).

## 3. Results and Discussion

### 3.1. Pollution Characteristics of TBBPA in the Weihe River Basin

#### 3.1.1. Pollution Characteristics of TBBPA in Sediment

The TBBPA concentrations in the sediment samples from the Weihe River Basin are shown in Table 1. The detection frequency of TBBPA in 34 sediments was 61.8%. The TBBPA concentrations ranged from not detected (N.D.) to 3.889 ng/g dw (dry weight), and the mean was 0.283 ng/g dw. The highest TBBPA concentration found in sediment was in the Yellow River (3.889 ng/g dw). The TBBPA concentrations in the sediment samples from different regions are shown in Table 2. Compared with TBBPA (dw) in sediment of rivers and lakes in other countries, the TBBPA (dw) in the Weihe River was significantly lower than that in St. Lawrence, Canada [13], and one order of magnitude lower than that in the Ebro river in Spain [14]. The TBBPA concentrations were similar to those in the six major lakes in the U.K. [15]. However, the TBBPA concentrations were slightly higher than that in the Scholte River in Netherlands [16] and Lake Mjosa in Norway [17]. In contrary to the other regions in China, the TBBPA (dw) in the sediments of Weihe River was significantly lower than that in Chaohu Lake [18] and one order of magnitude lower than that in Beijing Qinghe [19] and Erhai Lake [20]. These TBBPA concentrations were comparable to Taihu [21,22], Xijiang River, and Beijiang River [23]. This indicates that the Weihe River is relatively lightly contaminated with TBBPA [24].

#### 3.1.2. Pollution Characteristics of TBBPA in Water Sample

The TBBPA concentrations in the water samples from the Weihe River Basin are shown in Table 1. TBBPA was found in 36 (27.8%) of the samples. The TBBPA concentrations ranged from not detected (N.D.) to 12.279 ng/L, and the mean was 0.937 ng/L. The highest TBBPA concentration found in water was 12.279 ng/L after the Qingshui River merged into the Weihe River. The TBBPA concentrations in the water samples from different regions are showed in Table 2. The TBBPA concentrations in the Weihe River were significantly lower than the concentrations found in Chaohu Lake [18], and one order of magnitude lower than those in the Qinghe River in Beijing [25]. The TBBPA concentrations in the Weihe River were similar to concentrations that have been found in water from Taihu Lake [21,22] and Dongjiang River [26]. This indicates that the Weihe River is relatively lightly contaminated with TBBPA [24].

### 3.2. Spatial Distribution Characteristics and Source Analysis of TBBPA in the Weihe River Basin

#### 3.2.1. Spatial Distribution Characteristics of TBBPA in Sediments

The spatial distribution of TBBPA concentrations in sediment in the study area is shown in Figure 2A. The concentration distribution of TBBPA in the Weihe River Basin was downstream > upstream > middle stream. This may be due to the smaller particles in the downstream sediments, the slower water flow rate, and the enhanced adsorption of TBBPA by sediments, resulting in a significant residual concentration of TBBPA in the downstream sediments. The highest TBBPA concentration was founded in Yellow River sediment, possibly because this site is a tourist attraction with many water recreation facilities and nearby farmhouses, which may contain TBBPA. At the same time, the residual amount of TBBPA in the sediment of mainstream was higher than that in the tributaries, which may be closely related to the long-distance migration of TBBPA and its persistent, and difficult decomposition in the environment.

#### 3.2.2. Spatial Distribution Characteristics of TBBPA in Water Sample

The spatial distribution of TBBPA in water samples in the study area is shown in Figure 2B. The concentration distribution of TBBPA in the Weihe River Basin was downstream > upstream > middle stream. This may be due to the existence of a small number of electronics and textile industries in the upstream cities. In these industries, TBBPA is widely used as a flame retardant. During the use of TBBPA, it will enter the river water with the discharge of wastewater and though the migration and sedimentation process of the atmosphere [27]. Moreover, the density of the surrounding population leads to the increase of the use of products containing TBBPA. Therefore, the content of TBBPA in the upstream water samples was higher than that in the middle and downstream water samples. In addition, the concentration of TBBPA in the mainstream was much larger than that of the tributaries, which was determined by the specific properties of TBBPA. TBBPA has a long half-value period in the environment and is highly mobile with long-distance migration, resulting in a higher TBBPA concentration in the mainstream than in the tributary [17].

#### 3.2.3. Correlation between TBBPA and TOC in Sediments

Studies have shown that hydrophobic organic pollutants were closely related to the content of TOC [28,29], and the TOC content of sediment was one of the important factors affecting the retention, migration, and transformation of hydrophobic organic pollutants in sediments [30]. Wang Junxia [3] found that TBBPA was significantly correlated with TOC content in the research of soil in Sichuan and Tibet. To investigate the effect of TOC content on the distribution of TBBPA, the correlation of TBBPA with TOC in 34 sediment samples was analyzed (shown in Figure 3).

The correlation coefficient between TOC and TBBPA concentration was 0.00566 (R^2^). The results showed that there was no correlation between TOC and TBBPA in 34 sediments, which indicates that in addition to the influence of TOC, TBBPA in sediments may be affected by dense atmospheric and terrestrial inputs, wet deposition, remote transmission, etc. [31,32,33].

#### 3.2.4. Source Analysis

The results of this study showed that the content of TBBPA in the water samples of 9#, 10#, 26#, 28#, and 36# in the upstream of Weihe River were higher. This could be due to the existence of some electronics and textile industries in the upstream cities, and the dense population density in the region has led to an increase in the use of electronic products containing TBBPA. Therefore, TBBPA in sediment and water samples may come from the use of electronic products containing TBBPA, and the influence of the electronics industry and the textile industry, combined with regional characteristics and development characteristics.

### 3.3. Ecological Risk Assessment

#### 3.3.1. Screening, Acquisition, and Collation of Toxicological Data

Toxicological data for TBBPA were obtained from the U.S. Environmental Protection Agency’s EPA ECOTOX Toxicology Database, CNKI (China National Knowledge Infrastructure), and published literature. All data were filtered and classified according to the following principles. (1) The whole species are divided into three categories, algae, invertebrates (crustaceans, insects and arachnids, mollusks, worms, and others), vertebrates (fish, amphibians) [34]. Furthermore, the aquatic organisms (including local and introduced species) in the Weihe River Basin were filtered from the database. (2) The screening conditions for chronic toxicological numbers are shown in Table 3. The exposure time of chronic toxicological species data was at least 14 d for vertebrates, 7 d for scorpions (invertebrates), and 3 d for algae. The acute toxicological data (EC_50_, LC_50_) with exposure time within 96 h were selected. (3) The most sensitive data of test endpoints was selected when the same species had multiple sets of data. When there were multiple sets of different test data for the same species, the same test endpoint, and the same exposure time, the geometric mean of these data was calculated [35].

Generally, the chronic toxicological data of organism can reflect the exposure risk of organisms in the actual ecological environment better. Due to the complicated acquisition and long experiment cycle of chronic toxicological data, the final screened data had difficulty in meeting the requirement of the Technical Guidance Document on Risk Assessment (TGD) standard for aquatic organisms of eight families. Therefore, the acute and chronic rate method (ACR) was used to extrapolate the chronic toxicological data. However, the acute and chronic rate of the species in this study are difficult to obtain, so 10 was selected as the default value [36]. The acute and chronic toxicological data after screening are shown in Table 4.

#### 3.3.2. The Calculation of PNEC_wat_ in Water Body

The PNEC_wat_ is computed by constructing the SSD curve. The basic connotation of the SSD method is that the sensitivity of different species to a stress factor is subject to a certain (cumulative) probability distribution, and the sensitivity difference of different species samples to the stress factor can be described by probability or empirical distribution function in an ecosystem with complex structure [39]. The basic assumption of SSD is that the sensitivity of a group of organisms can be subject to some distribution. The commonly used fitting functions of SSD are (log-logistic), normal-distributed (log-normal) and log-triangular, and the commonly used fitting functions in Europe and America mainly include logistic functions (logistic, sigmoid, Weibull, etc.), log-positive distribution functions (such as lognormal, normal, etc.), and trigonometric functions [37].

The selected toxicological data was used to fit the curve. In the construction of the SSD model, the selected species involved three trophic levels and eight families, and the total amount of data was 11. The amount of data required to estimate the SSD curve is different for different standards. TGD [40] requires at least 8 different families of organism and 10 sets of chronic data, the U.S. EPA [41] requires at least 3 phyla and 8 families of aquatic organism and above one aquatic plant, and the OECD [42] requires at least 5 sets of NOEC or MTC (maximum tolerance concentration) of different species. Therefore, the data screened in this study meets the requirements. Then the SSD curve was constructed using the selected data of aquatic organisms, and the curve fitting was performed using SigmaPlot 14.0 software. The software was more flexible, and its model can be selected from log-normal, log-logistic, sigmoid, Weibull, Gompertz, and other built-in functions. Based on the value of R^2^ and the fitting curve, the toxicity value HC_5_ was inferred. In the model building process, a variety of fitting functions were used for fitting, including Gaussian model (3 parameters), exponential growth model (3 parameters), Weibull (4 parameters), and sigmoid (3 parameters). The fitting results are shown in Table 5. According to multiple fitting and comparative analysis, the sigmoid model has an R^2^ value of 0.9768, showing that the degree of fitting is high. The value of HC_5_ was 6.74 μg/L. When the amount of chronic toxicity data is sufficient, the value of HC_5_ derived from the SSD model is PNEC. When the amount of chronic toxicity data is insufficient, the ratio of HC_5_ to AF (1~5) is used as the final PNEC value. Considering the worst situation, 5 was selected as the value of AF. Therefore, the value of PNEC_wat_ is 1.35 μg/L.

#### 3.3.3. The Calculation of PNEC_sed_ in Sediments

There are few toxicological data on TBBPA in sediments for literature, leading to difficulties in obtaining the value of PNEC_sed_ [43]. Therefore, The PNEC_sed_ was calculated by using the equilibrium distribution method (TGD) [44].
(1)$$PNEC_{sed\;dry}=\frac{K_{susp-water}}{RHO_{susp}}\times PNEC_{water}\times 1000\times 4.6$$
(2)$${RHO}_{susp}=F_{solid-susp}\cdot RHO_{solid}+F_{water-susp}\cdot RHO_{water}$$
(3)$$K_{susp-water}=F_{water-susp}+F_{solid-susp}\times F_{oc-susp}\times \frac{K_{oc}}{1000}\times RHO_{solid}$$
where $RHO_{susp}$ is suspension wet volume density, $RHO_{solid}$ is solid phase density, $\;RHO_{water}$ is water density, $K_{susp-water}$ is suspended matter–water distribution coefficient, $F_{water-susp}$ is the volume fractions of water in suspension, $F_{solid-susp}$ is the volume fractions of solid in suspension, Koc is pollutant organic carbon–water partition coefficient. Koc value of 31,027.96 was taken from previous publications [45]. The value of ${PNEC}_{sed}$ was 6.14 ng/L. The default value of $F_{solid-susp}$ was 0.1 m^3^/m^3^, the default value of ${RHO}_{susp}$ was 2500 kg/m^3^, the default value of $F_{water-susp}$ was 0.9 m^3^/m^3^, the default value of $RHO_{water}$ was 1000 kg/m^3^, and the default value of $F_{oc-susp}$ was 0.1 kg/kg. By calculation, the value of PNEC_sed_ is 422.98 ng/g.

#### 3.3.4. Risk Assessment

Risk quotients (RQs) are frequently adopted as a practical tool to characterize the ecological risk of organics on aquatic organisms in river [46,47]. Therefore, the method of RQ was used to evaluate the ecological risk of TBBPA in this study. The quotient method is based on the integration process of limited biotoxicity data and exposure data for natural water pollutants. Generally, different levels are classified by the results of the actual exposure model and the exposure concentration that characterizes the risk level of the substance to characterize the magnitude of the ecosystem risk. The specific calculation method is
(4)RQ=MEC/PNEC
where RQ is risk quotient and MEC is measured environmental concentration. Three risk levels were used based on the RQ. Generally speaking, RQ > 1 was classed as high risk, 0.1 < RQ < 1 as medium risk, and RQ < 0.1 as low risk [48]. The RQ range of 36 water samples was 5.30 × 10^−4^~9.01 × 10^−3^, and the mean value of RQ was 2.44 × 10^3^. As for sediment samples, the RQ range was 5.67 × 10^−8^~2.28 × 10^−5^ and the mean value of RQ was 1.08 × 10^−6^. In the Weihe River Basin, the RQ of water and sediments was below 0.01, indicating that TBBPA has no/low risk for aquatic organisms. Moreover, the risks of TBBPA in the sediment were significantly lower than those in the water. Therefore, the risk assessment in water required more attention.

## 4. Conclusions

This paper studied the TBBPA in sediments and waters in the Weihe area of Shaanxi Province. The detection frequency of TBBPA in sediments and water samples was 61.8% and 27.8%, respectively; the TBBPA concentrations in sediments and water samples were in the range of N.D.–3.889 ng/g (mean value of 0.283 ng/g) and N.D. –12.279 ng/L (mean value of 0.937 ng/L), respectively. The results indicated that the Weihe River is relatively lightly contaminated with TBBPA. The spatial distributions of TBBPA in surface sediments and water indicated that the local point-input was their major source. TBBPA in sediments and water samples may originate from the use of electronic products containing TBBPA and the influence of the electronics industry and the textile industry.

There was no correlation between TBBPA and TOC in sediment samples, indicating that TBBPA in sediments may be affected by other factors besides being affected by TOC. The potential risks posed by TBBPA in water and sediment were characterized using the risk quotient method. The risk quotient of TBBPA at each sampling point in water and sediments was calculated, showing that the RQ of TBBPA was less than 0.01. The results indicated that TBBPA has no or low potential risks in water and sediments on aquatic organisms.

## Figures and Tables

**Figure 1 ijerph-17-03750-f001:**
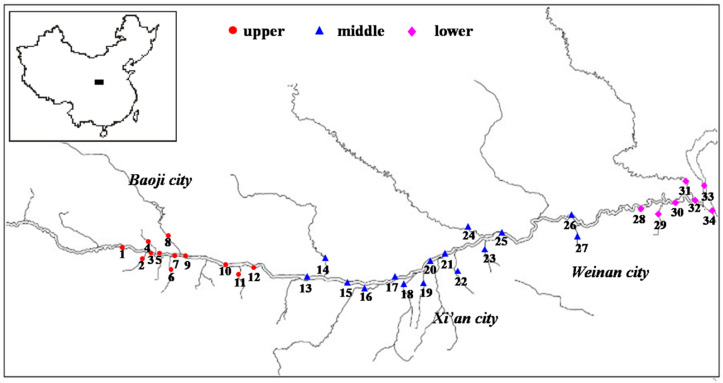
Sampling location of surface sediments in Weihe River Basin Shaanxi section, China [12].

**Figure 2 ijerph-17-03750-f002:**
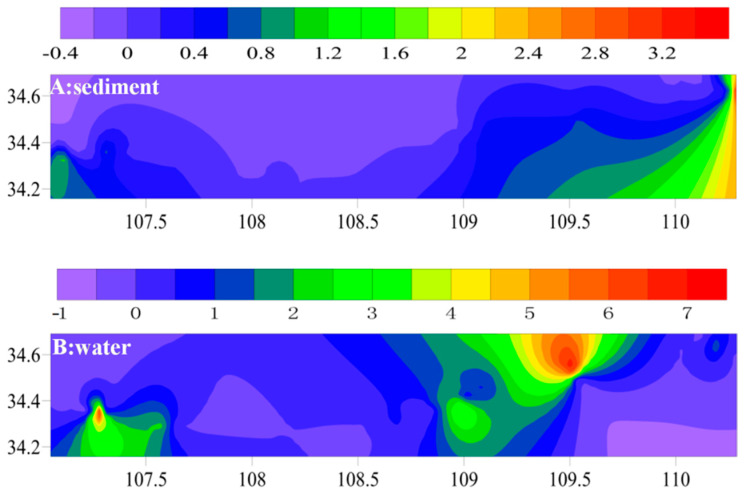
Spatial distribution of TBBPA in sediment (**A**) and water (**B**).

**Figure 3 ijerph-17-03750-f003:**
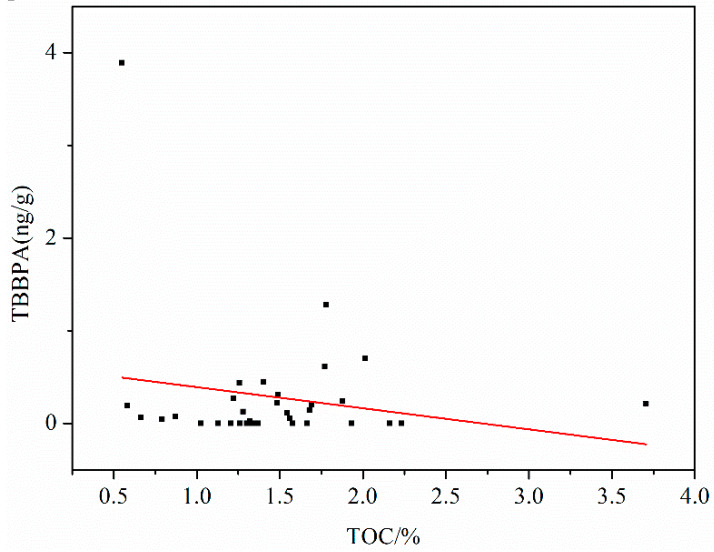
Correlation between TBBPA and TOC in sediment samples of the Weihe River Basin.

**Table 1 ijerph-17-03750-t001:** Residue levels of tetrabromobisphenol A (TBBPA) in the Weihe River.

Zones	Number	Types	Sediment		Water Sample
			TOC/%	TBBPA (ng/g, dw)	TBBPA (ng/L)
Upper stream	1	Main stream	1.328	N.D.	N.D.
	2	Tributary stream	1.779	1.281	N.D.
	3	Main stream	1.544	0.111	N.D.
	4	Tributary stream	2.163	N.D.	N.D.
	5	Main stream	1.667	N.D.	N.D.
	6	Tributary stream	2.016	0.700	N.D.
	7	Main stream	0.584	0.190	1.295
	8	Tributary stream	1.279	0.127	N.D.
	9	Main stream	1.32	0.024	12.279
	10	Main stream	1.691	0.203	2.137
	11	Tributary stream	3.707	0.213	N.D.
	12	Main stream	1.931	N.D.	N.D.
Middle stream	13	Main stream	1.34	N.D.	N.D.
	14	Tributary stream	0.87	0.069	N.D.
	15	Main stream	1.576	N.D.	N.D.
	16	Tributary stream	0.665	0.062	N.D.
	17	Main stream	0.792	0.043	N.D.
	18	Tributary stream	2.234	N.D.	N.D.
	19	Tributary stream	1.301	N.D.	N.D.
	20	Main stream	1.562	0.052	0.716
	21	Tributary stream	-	-	N.D.
	22	Main stream	1.025	N.D.	1.377
	23	Tributary stream	-	-	N.D.
	24	Main stream	1.207	N.D.	1.493
	25	Tributary stream	1.257	0.432	N.D.
	26	Tributary stream	1.483	0.221	4.086
	27	Main stream	1.402	0.450	0.890
Lower stream	28	Main stream	1.491	0.309	7.203
	29	Tributary stream	1.772	0.614	N.D.
	30	Main stream	1.369	N.D.	N.D.
	31	Tributary stream	1.678	0.140	N.D.
	32	Main stream	1.261	N.D.	N.D.
	33	Tributary stream	1.128	N.D.	N.D.
	34	Main stream	1.219	0.265	1.523
	35	Main stream	1.878	0.241	N.D.
	36	Main stream	0.55	3.889	N.D.

N.D: not detected. TOC: total organic carbon.

**Table 2 ijerph-17-03750-t002:** The comparison of TBBPA content in sediment and water from home and abroad.

Sample	Sample Collection Area	Sample Type	TBBPA/(ng/g)	References
	China			
	Chaohu Lake	Lake sediments	Max. 518	[5]
	Taihu Lake	Lake sediments	0.056–2.15	[21]
	Xijiang River	River bottom sediments	N.D. –1.33	[23]
	Beijiang River	River bottom sediments	0.537–6.2	[23]
	Beijing Qinghe	River bottom sediments	0.2–22	[19]
Sediments	Erhai Lake	Lake sediments	21–53	[20]
	Other countries			
	Netherlands Scholt River	River bottom sediments	<0.1	[16]
	Canada St. Lawrence River	River bottom sediments	300	[13]
	A River in Paris, France	Lake sediments	0.04–0.13	[17]
	Britain Six major Lakes	Lake sediments	0.33–3.8	[15]
	Spain Ebro River	River bottom sediments	N.D. –15	[14]
	China			
Water	Taihu Lake	Lake	N.D. –1.12	[21]
	Beijing Qinghe	River	23.9–224	[25]
	Chaohu Lake	Lake	850–4870	[18]
	Dongjiang River	River	1.11–2.83	[26]
	Other countries			
	England	Lake	0.14–3.2	[15]
	France	River	<0.035–0.068	[17]

N.D: not detected.

**Table 3 ijerph-17-03750-t003:** Screening conditions of chronic toxicological data.

Species	Observation Endpoint	Exposure Time/d	Water Type	Experimental Site
Algae	NOEC/LOEC	≥3	Fresh water	Laboratory test
Invertebrate	NOEC/LOEC	≥7	Fresh water	Laboratory test
Vertebrate	NOEC/LOEC	≥14	Fresh water	Laboratory test

**Table 4 ijerph-17-03750-t004:** Chronic toxicological data of TBBPA.

The Most Sensitive Species	Biological Classification	Endpoint	Standard Concentration mg/L	Data Source
*Chlorella pyrenoidosa*	Chlorophyta Chlorellaceae *Chlorella*	LOEC	2.67	ECOTOX
*Scenedesmus acutus* var. *acutus*	Chlorophyta Scenedesmaceae *Scenedesmus*	NOEC	0.50	ECOTOX
*Daphnia magna*	Arthropoda Daphniidae *Daphnia*	NOEC	1.80	ECOTOX
*Carassius auratus*	Chordata Cyprinidae *Carassius*	NOEC	0.28	ECOTOX
*Gobiocypris rarus*	Chordata Cyprinidae *Gobiocypris*	NOEC	0.05	[37]
*Brachionus calyciflorus Pallas*	Rotifera Brachionidae *Brachionus*	NOEC	1.00	[37]
*Brachydanio rerio*	Chordata Cyprinidae *Danio*	EC_50_	22.00	[38]
*Gammarus pulex*	Arthropoda Gammaridae *Gammarus*	LC_50_	1.17	ECOTOX
*Pseudorasbora parva*	Chordata Cyprinidae *Pseudorasbora*	LC_50_	0.86	ECOTOX
*Limnodrilus hoffmeisteri*	Annelida Tubificidae *Limnodrilus*	EC_50_	2.92	ECOTOX
*Chironomus plumosus*	Arthropoda Chironomidae *Chironomus*	LC_50_	0.13	ECOTOX

**Table 5 ijerph-17-03750-t005:** Fitting results.

Model	Equation	R^2^	Standard Error of Estimate
Exponential Growth	Y = 0.70 + 315.57 × exp (0.0005 × x)	0.9674	0.056
Gaussian	Y = 0.88 × exp (−0.5 × ((x − 1.73/2.81^2^)	0.9752	0.049
Weibull	Y= 1.00 × (1 − exp (−(abs(x + 1.24 + 6.80 × ln(2)^(1 / 3.17)) / 6.80)^3.17)))	0.9765	0.052
Sigmoid	Y = 0.98 / (1 + exp (−(x + 1.33) / 1.26))	0.9768	0.048

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
