# Peer review of "Contamination Level, Distribution Characteristics, and Ecotoxicity of Tetrabromobisphenol A in Water and Sediment from Weihe River Basin, China"

_ijerph, 2020, doi:10.3390/ijerph17113750_

Round 1

Reviewer 1 Report

This paper inspected the concentration and distribution of Tetrabromobisphenol (TBBPA) concentrations in water and sediment samples collected from the Weihe River Basin (China). Moreover, this research examined the relationship between TBBPA and the organic content of the river. Ultimately, the potential risk of this contaminant in the sediment and water was determined. The manuscript is very well structured and written. The analytical method has been correctly applied and explained, and the results are clear and well discussed. Overall, it has been observed that the concentration of TBBPA in sediment in the Weihe River Basin has a potential risk at two sites and that the organic matter content of the river has an insignificant effect.

This reviewer has only minor comments:

In general, more attention should be paid to the significance of the study in the introduction section. Line 138: I think water samples, not sediment, please check

Line 139: please check the units

Line 168: This sentence needs to be revised in line with the figure

Line 289: please correct the English mistake

Author Response

Response to Reviewer 1 Comments

Thanks for all your good suggestions and we think it is a really essential and worth considering comment. According to your suggestions, we have carefully revised our manuscript, especially include our language, discussion and conclusion. The specific comments mentioned was revised in the following (language errors are not listed):

Point 1:

1.1. In general, more attention should be paid to the significance of the study in the introduction section.

1.2.Line 138: I think water samples, not sediment, please check

Response 1: Thank you very much for your suggestion. As suggestion, the text has been revised. The revised words are as follows:

TetrabromobisphenolA (TBBPA) is the most widely used brominated flame retardant in the world, accounting for more than 60% of the brominated flameretardant market, and is extensively used in printed circuit board, paper, textile and other industrial products [1]. It was demonstrated in studies that TBBPA has strong immunotoxicity, endocrine toxicity and the characteristics of persistent pollutants which are the long-distance migration, bioaccumulation and toxicity[2-3]. TBBPA continues to enter the environment in the process of production, use and disposal of related products. Therefore, the harm of TBBPA to the environment and human body has been widely concerned, and it has been detected in various biological and environmental matrices including soil, atmosphere, water, sediment, animals and plants[4-7].

Recent studies indicate that there are a number of pathways through which TBBPA can enter the environment, including release from consumer products, emissions from production processes, and leaching from treatment sites. As a reactive flame retardant, TBBPA does not migrate from the product. However, as an additive flame retardant, TBBPA can be released from the product and enter indoor air and dust[8]. When TBBPA was used as an additive flame retardant, it was mainly used in the outer casing of electronic products such as televisions and computers. It has been found that the main sources of TBBPA in indoor air and dust were old electronic devices and electrical appliances, especially televisions, computers, etc[9]. Therefore, in places where the population density is high, and the activity area is strong, the large-scale use of electronic products containing TBBPA is the main reason for the increase in the concentration of TBBPA in the environment.

Weihe River, the largest tributary of the Yellow River, mainly flows through Tianshui, Baoji, Xianyang, Xi'an and Weinan. Millions of people were living along the Weihe River basin, and there are many chemical, electronics manufacturing, and textile enterprises in the Weihe River Basin. Rapid population increases and economic development have led to large quantities of anthropogenic contaminants entering the Weihe River from primary sources in runoff, in industrial and domestic effluent, and through atmospheric deposition. Previous studies have reported that organic-chlorine pesticides[10] and polycyclic aromatic hydrocarbons[11] were detected in environmental media of Weihe River Basin. However, few studies have been focused on flame retardant pollutants, and the related research on TBBPA has not yet been carried out. Therefore, investigation and research on new POPs such as TBBPA has become very urgent and necessary. In this study, we selected the Weihe River Basin of the Shaanxi section as the research area to analyze the concentration level, spatial distribution characteristics and sources of TBBPA in sediment and suspended solids, which can provide the basic data for the assessment of environmental occurrence in the Weihe River Basin. The obtained results could provide data support for the global inventory of TBBPA, and provide data information for the future work on risk assessment and pollution control. (Line 34-68)

Due to my negligence, I made a mistake and have made correction in the revised manuscript. The “sediment samples” has been changed to “water samples”.

Point 2:

Line 139: please check the units

Response 2: Thank you very much for your reminding. It’s my careless, the unit “ng/g dw” has been changed to “ng/L” in the revised manuscript.

Point 3:

Line 168: This sentence needs to be revised in line with the figure

Response 2: Thank you very much for your suggestion. As suggestion, the text has been revised. The sentence “The TBBPA concentrations in downstream were comparable to middle stream.” has been deleted.

Point 4:

Line 289: please correct the English mistake

Response 3: Thank you very much for your correction. As suggestion, the text has been revised as follows: “The results indicated that the Weihe River is relatively lightly contaminated with TBBPA”.

Reviewer 2 Report

In this study, environmental water and sediment samples of the Weihe river basin, located in China, were investigated for brominated flame retardant tetrabromobisphenol A (TBBPA) contamination, particularly focusing on its spatial distribution characteristics and risk assessment modeling. It is recognized a regional interest and relevance of the subject. Although relatively satisfactory written, some parts were identified of being confusing in their message. Reading is relatively easy to follow. A few references are lacking as support of statements. In terms of experimental methodology and results some concerns were highlighted. Discussions are at times not supported by current data, and conclusions are confusing and superficial, taking into account the specific thematic of this manuscript. Revision by a native English speaker is recommended. Figures and Tables require minor adjustments. Detailed questions/concerns were highlighted through the manuscript document enclosed - please see attached.

Author Response

Response to Reviewer 2 Comments

Thanks for all your good suggestions and we think it is a really essential and worth considering comment. According to your suggestions, we have carefully revised our manuscript; especially include our language, discussion and conclusion.

In this study, environmental water and sediment samples of the Weihe river basin, located in China, were investigated for brominated flame retardant tetrabromobisphenol A (TBBPA) contamination, particularly focusing on its spatial distribution characteristics and risk assessment modeling. It is recognized a regional interest and relevance of the subject. Although relatively satisfactory written, some parts were identified of being confusing in their message. Reading is relatively easy to follow. A few references are lacking as support of statements. In terms of experimental methodology and results some concerns were highlighted. Discussions are at times not supported by current data, and conclusions are confusing and superficial, taking into account the specific thematic of this manuscript. Revision by a native English speaker is recommended. Figures and Tables require minor adjustments. Detailed questions/concerns were highlighted through the manuscript document enclosed - please see attached.

Response: Thank you very much for your suggestion. As suggestion, some new references have been added, such as reference21,23,and 31.

Discussions and conclusions section have been modified. The revised conclusion as follows: This paper studied the TBBPA in sediments and waters in the Weihe area of Shaanxi Province. The detection frequency of TBBPA in sediments and water samples were 61.8% and 27.8%, respectively; the TBBPA concentrations in sediments and water samples were in the range of N.D.-3.889 ng/g (mean value of 0.283 ng/g) and N.D.-12.279 ng/L (mean value of 0.937 ng/L), respectively. The results indicated that the Weihe River is relatively lightly contaminated with TBBPA. The spatial distributions of TBBPA in surface sediments and water indicated that the local point-input was their major source. TBBPA in sediments and water samples may be originated from the use of electronic products containing TBBPA, and the influence of the electronics industry and the textile industry.

There was no correlation between TBBPA and TOC in sediment samples, indicating that TBBPA in sediments may be affected by other factors besides being affected by TOC. The potential risks posed by TBBPA in water and sediment were characterized using the risk quotient method. The risk quotient of TBBPA at each sampling point in water and sediments in the Weihe River was calculated, showing that the RQ of TBBPA at some sampling points were greater than 1. It is indicated that TBBPA has potential risks in water and sediments in certain areas.(Line 292-306)

The revised manuscript has been revised by a native English speaker. The manuscript has made a lot of changes, so here is not list one by one, please refer to the revised draft for details.

Reviewer 3 Report

The paper is clear and well written.

The topic appropriate for the Journal.

The paper was correctly edited and graphically developed at a good level.

I do not see any shortcuts.

The paper deserves a positive assessment because it is current and interesting from both a cognitive and practical point of view.

But the introduction chapter, the citation of the literature in the paper, and the literature chapter should be improved.

Author Response

Response to Reviewer 3 Comments

Thank you very much for your recognition and support. And Thanks for all your good suggestions and we think it is a really essential and worth considering comment. According to your suggestions, we have carefully revised our manuscript, especially include language, figures, introduction chapter, results and the citation of the literature in the paper. At the same time, the literature chapter has been improved.

Point 1: Comments and Suggestions for Authors

But the introduction chapter, the citation of the literature in the paper, and the literature chapter should be improved

Response 1: Thank you very much for your suggestion. As suggestion, the introduction chapter been revised. The revised words are as follows:

TetrabromobisphenolA (TBBPA) is the most widely used brominated flame retardant in the world, accounting for more than 60% of the brominated flameretardant market, and is extensively used in printed circuit board, paper, textile and other industrial products [1]. It was demonstrated in studies that TBBPA has strong immunotoxicity, endocrine toxicity and the characteristics of persistent pollutants which are the long-distance migration, bioaccumulation and toxicity[2-3]. TBBPA continues to enter the environment in the process of production, use and disposal of related products. Therefore, the harm of TBBPA to the environment and human body has been widely concerned, and it has been detected in various biological and environmental matrices including soil, atmosphere, water, sediment, animals and plants[4-7].

Recent studies indicate that there are a number of pathways through which TBBPA can enter the environment, including release from consumer products, emissions from production processes, and leaching from treatment sites. As a reactive flame retardant, TBBPA does not migrate from the product. However, as an additive flame retardant, TBBPA can be released from the product and enter indoor air and dust[8]. When TBBPA was used as an additive flame retardant, it was mainly used in the outer casing of electronic products such as televisions and computers. It has been found that the main sources of TBBPA in indoor air and dust were old electronic devices and electrical appliances, especially televisions, computers, etc[9]. Therefore, in places where the population density is high, and the activity area is strong, the large-scale use of electronic products containing TBBPA is the main reason for the increase in the concentration of TBBPA in the environment.

Weihe River, the largest tributary of the Yellow River, mainly flows through Tianshui, Baoji, Xianyang, Xi'an and Weinan. Millions of people were living along the Weihe River basin, and there are many chemical, electronics manufacturing, and textile enterprises in the Weihe River Basin. Rapid population increases and economic development have led to large quantities of anthropogenic contaminants entering the Weihe River from primary sources in runoff, in industrial and domestic effluent, and through atmospheric deposition. Previous studies have reported that organic-chlorine pesticides[10] and polycyclic aromatic hydrocarbons[11] were detected in environmental media of Weihe River Basin. However, few studies have been focused on flame retardant pollutants, and the related research on TBBPA has not yet been carried out. Therefore, investigation and research on new POPs such as TBBPA has become very urgent and necessary. In this study, we selected the Weihe River Basin of the Shaanxi section as the research area to analyze the concentration level, spatial distribution characteristics and sources of TBBPA in sediment and suspended solids, which can provide the basic data for the assessment of environmental occurrence in the Weihe River Basin. The obtained results could provide data support for the global inventory of TBBPA, and provide data information for the future work on risk assessment and pollution control.( Line 34-68)

The citation of the literature in the paper was changed to numerical format, and the literature chapter has also been improved according to the change of the citation in the paper.

Reviewer 4 Report

The research results presented by the authors in “Contamination Level, Distribution Characteristics, and Ecotoxicity of Tetrabromobisphenol A in Water and Sediment from Weihe River Basin, China” are interesting, however not very useful for readers from different part of the world. I  have several comments regarding mainly the methods and results presentation, which would make it more understandable and valuable for more readers. I believe that my detail comments to the authors will be useful.

Abstract

It gives the overall view on the results. The aim of the research is not clearly stated. Same of the conclusion driven from the research are not given here, eg. concerning the impact of development characteristics.

Introduction

Move here same general information from Results (detailes below). Add also general information about ecotoxicity from 3.3.1. At the end of introduction part, state clearly the aim of the research.

The description of the study site should be a separate point of Materials and Methods. Line 43-44 - provide more detailed information about types of industries and agriculture, as well as population density in the region of Weihe River.

Materials and Methods

This part should contain 3 issue: Description of the study site, Methods of laboratory test, Methods of Ecological Risk Assessment. The methodology of laboratory tests is described in too much detail.

Results.

Please find different way of result presentation. Three big tables - one by one do not encourage to pay attention on it.

Figure 3 – What does mean: “The correlation between TOC and TBBPA is 0,00566”. This value is very low so it is difficult to deduce a correlation,  especially without giving R2.

Line 212 - Information on the types of industries in the region should be included in the introduction, in addition to information on which industries may affect TBBPA concentrations in the water. From the results of the research it can be concluded which of them influence the TBBPA concentration in the region.

Line 198-208 – move to the introduction

Conclusions

A few more conclusions were made on the basis of the research. They are given somewhere in the paper, but you should repeat them here.

Author Response

Response to Reviewer 4 Comments

Thanks for all your good suggestions and we think it is a really essential and worth considering comment. According to your suggestions, we have carefully revised our manuscript; especially include our language, materials and methods, discussion and conclusion. The specific comments mentioned were revised in the following:

Point 1: Abstract

It gives the overall view on the results. The aim of the research is not clearly stated. Same of the conclusion driven from the research are not given here, eg. concerning the impact of development characteristics.

Response 1: Thank you very much for your suggestion. As suggestion, the abstract chapter been revised. The revised words are as follows (The focus section has been highlighted in bold) : Tetrabromobisphenol A (TBBPA) is a brominated flame retardant, which is widely present in the various environmental and biological media. The knowledge on the contamination of TBBPA in Weihe River Basin is still limited. In order to know the pollution level and distributions of Tetrabromobisphenol A (TBBPA) in the Weihe River Basin, a total of 34 sediment samples and 36 water samples were collected from main stream and tributary, and the concentration of TBBPA in the samples were analyzed by high performance liquid chromatography-electrospray ionization-mass spectrometry (HPLC-ESI-MS). The detection frequency of TBBPA in sediments and water samples were 61.8% and 27.8%, respectively; the TBBPA concentrations in sediments and water samples were in the range of N.D.-3.889 ng/g (mean value of 0.283ng/g) and N.D.-12.279 ng/L (mean value of 0.937ng/L), respectively. Compared with other areas in China, the residues of TBBPA in the Weihe River Basin were in a relatively low level. The spatial distributions of TBBPA in surface sediments and water indicated that the local point-input was their major source. This is related to the proximity of some sampling sites to industrial areas and domestic sewage discharge areas. The insignificant correlation between TBBPA and total organic carbon (TOC) indicated that TBBPA in sediments is not only influenced by TOC, but also affected by atmosphere and land input, wet deposition and long-distance transmission. The potential risks posed by TBBPA in water and sediment were characterized using the risk quotient method. The RQs of TBBPA at some sampling points were greater than 1, indicating that TBBPA has potential risks in water and sediments in certain areas.(Line 11-29)

Point 2:  Introduction

Move here same general information from Results (detailes below). Add also general information about ecotoxicity from 3.3.1. At the end of introduction part, state clearly the aim of the research.

The description of the study site should be a separate point of Materials and Methods. Line 43-44 – provide more detailed information about types of industries and agriculture, as well as population density in the region of Weihe River.

Response 2: Thank you very much for your suggestion. As suggestion, the abstract chapter been revised. The revised words are as follows: Weihe River, the largest tributary of the Yellow River, mainly flows through Tianshui, Baoji, Xianyang, Xi'an and Weinan. Millions of people were living along the Weihe River basin, and there are many chemical, electronics manufacturing, and textile enterprises in the Weihe River Basin. Rapid population increases and economic development have led to large quantities of anthropogenic contaminants entering the Weihe River from primary sources in runoff, in industrial and domestic effluent, and through atmospheric deposition. Previous studies have reported that organic-chlorine pesticides[10] and polycyclic aromatic hydrocarbons[11] were detected in environmental media of Weihe River Basin. However, few studies have been focused on flame retardant pollutants, and the related research on TBBPA has not yet been carried out. Therefore, investigation and research on new POPs such as TBBPA has become very urgent and necessary. In this study, we selected the Weihe River Basin of the Shaanxi section as the research area to analyze the concentration level, spatial distribution characteristics and sources of TBBPA in sediment and suspended solids, which can provide the basic data for the assessment of environmental occurrence in the Weihe River Basin. The obtained results could provide data support for the global inventory of TBBPA, and provide data information for the future work on risk assessment and pollution control.(Line 53-68)

Writing the study site in the abstract section can better reflect the research significance and value of the paper, so it is not described in the materials and methods section alone.

Point 3: Materials and Methods

This part should contain 3 issue: Description of the study site, Methods of laboratory test, Methods of Ecological Risk Assessment. The methodology of laboratory tests is described in too much detail.

Response 3: Thank you very much for your suggestion. The study site has been descripted in the abstract, so it can’t be repeated in this section. The method of ecological risk assessment has been added. The revised words are as follows: Toxicological data for TBBPA were obtained from the US Environmental Protection Agency's EPA ECOTOX Toxicology Database. The ecological risks presented by biotoxicity in the aquatic environment were evaluated using the risk quotient (RQ). The individual RQ was calculated as a ratio of the measured environmental concentration (MEC) and predicted no-effect concentration (PNEC) (RQ = MEC/PNEC). The PNEC is computed by constructing the SSD curve. The SSD curve was constructed using chronic toxicological data of aquatic organisms, and the curve fitting was performed using Sigmaplot14.0 software. (Line 135-141)

Point 4: Results.

Please find different way of result presentation. Three big tables - one by one do not encourage to pay attention on it.

Figure 3 – What does mean: “The correlation between TOC and TBBPA is 0,00566”. This value is very low so it is difficult to deduce a correlation, especially without giving R2.

Line 212 - Information on the types of industries in the region should be included in the introduction, in addition to information on which industries may affect TBBPA concentrations in the water. From the results of the research it can be concluded which of them influence the TBBPA concentration in the region.

Line 198-208 – move to the introduction

Response 4: Thank you very much for your suggestion. As suggestion, the table contented in the manuscript has been adjusted.

In Figure 3, “The correlation between TOC and TBBPA is 0.00566(R2)”. There is no doubt that the correlation is very low. Therefore, this conclusion is given in the paper “The results showed that there was no correlation between TOC and TBBPA in 34 sediments, which indicates that in addition to the influence of TOC, TBBPA in sediments may be affected by dense atmospheric and terrestrial inputs, wet deposition, remote transmission, etc.”

The information on the types of industries in the region has been included in the introduction, please refer to Response 2. The industry which effect TBBPA concentrations in the water has been given in the 3.2.2. Spatial distribution characteristics of TBBPA in water sample, listed as: This might be due to the existence of a small number of electronics industry and textile industry in the upstream cities.

The content of Line 198-208 has been move to the introduction

Point 5: Conclusions

A few more conclusions were made on the basis of the research. They are given somewhere in the paper, but you should repeat them here

Response 5: We have revised the conclusion part, the revised text as follows: This paper studied the TBBPA in sediments and waters in the Weihe area of Shaanxi Province. The detection frequency of TBBPA in sediments and water samples were 61.8% and 27.8%, respectively; the TBBPA concentrations in sediments and water samples were in the range of N.D.-3.889 ng/g (mean value of 0.283ng/g) and N.D.-12.279 ng/g (mean value of 0.937ng/g), respectively. The results indicated that the Weihe River is relatively lightly contaminated with TBBPA. The spatial distributions of TBBPA in surface sediments and water indicated that the local point-input was their major source. TBBPA in sediments and water samples may be originated from the use of electronic products containing TBBPA and the influence of the electronics industry and the textile industry.

There was no correlation between TBBPA and TOC in sediment samples, indicating that TBBPA in sediments may be affected by other factors besides being affected by TOC. The potential risks posed by TBBPA in water and sediment were characterized using the risk quotient method. The risk quotient of TBBPA at each sampling point in water and sediments in the Weihe River was calculated, showing that the RQ of TBBPA at some sampling points were greater than 1. It is indicated that TBBPA has potential risks in water and sediments in certain areas. (Line 292-306)

Round 2

Reviewer 2 Report

In this study, environmental water and sediment samples of the Weihe river basin, located in China, were investigated for brominated flame retardant tetrabromobisphenol A (TBBPA) contamination, particularly focusing on its spatial distribution characteristics and risk assessment modeling. It is recognized a regional interest and relevance of the subject. Although reading is relatively easy to follow, and despite adjustments were considered at the manuscript, some parts of the text remains confusing in their message. A few references are still lacking as support of statements. In terms of experimental methodology and results some concerns were highlighted, and not all were considered accordingly. Discussions were relatively improved, but conclusions remain confusing and superficial, taking into account the specific thematic of this manuscript. Revision by a native English speaker remains recommended. Figures and Tables require adjustments as previously highlighted that were not still clarified. Detailed questions/concerns were highlighted through the manuscript document enclosed - please see attached.
